# Microporous Block Copolymers Modified with Cu(II)-Coordinated Polyethylene Oxide-Substituted Silicas for Analytical Sensors

**DOI:** 10.3390/ma16206810

**Published:** 2023-10-22

**Authors:** Ilsiya M. Davletbaeva, Ekaterina D. Li, Zulfiya Z. Faizulina, Oleg O. Sazonov, Oleg V. Mikhailov, Karim R. Safiullin, Ruslan S. Davletbaev

**Affiliations:** 1Technology of Synthetic Rubber Department, Kazan National Research Technological University, Karl Marx str., 68, 420015 Kazan, Russia; katystayls@gmail.com (E.D.L.); faizulina.alin@yandex.ru (Z.Z.F.); sazonov.oleg2010@gmail.com (O.O.S.); karimsafiullin01@yandex.ru (K.R.S.); 2Material Science and Technology of Materials Department, Kazan State Power Engineering University, Krasnoselskaya str., 51, 420066 Kazan, Russia; darus@rambler.ru

**Keywords:** block copolymers, modification, Cu(II)-coordinated polyoxyethylene-substituted silicas, sorption activity, analytical test systems

## Abstract

The influence of stable-to-self-condensation Cu(II)-coordinated polyoxyethylene-substituted silicas (ASiP-Cu-0.5) on the synthesis of microporous block copolymers (OBCs) whose structural feature is the existence of coplanar polyisocyanate blocks of acetal nature (O-polyisocyanates) and a flexible-chain component of amphiphilic nature was studied. The use of ASiP-Cu-0.5 increased the yield of O-polyisocyanate blocks and the microphase separation of OBC. The resulting OBCs turned out to be effective sorbents for the analytical reagents PAN and PHENAZO, which, being in the micropore cavity, interacted with copper(II) and magnesium ions. To reduce the thickness of the selective OBC layer ten-fold and simplify the technology for obtaining analytical test systems, polyethylene terephthalate was used as a substrate for applying OBC. It was found that the increased sensitivity of the resulting test systems was due to the fact that in thin reaction layers, the efficiency of the formation of O-polyisocyanate blocks noticeably increased.

## 1. Introduction

Sensory membrane materials are widely used to determine cations and anions [1,2]. The structure and morphology of the membrane are the basis that determines the rapid ion exchange at the membrane/solution interface [3,4,5,6,7,8,9,10]. Block copolymers (BC) are promising as sensory membranes, which, being formed by two or more homopolymer fragments connected by covalent bonds, represent a specific category among polymeric materials. In contrast to mixtures of homopolymers, chemically different blocks in the composition of BC are forced to form intermolecular structures, a feature of which is microphase separation either in the solid state or in solution [11,12]. An important feature of BC is their ability to form a wide variety of supramolecular structures [13,14,15,16,17,18]. The order and size of such supramolecular formations are determined by the chemical structure, molecular weight, segregation ability of the block components, and the reaction conditions. [19,20,21,22,23]. The main factor affecting the morphology and degree of the long-range order of BCs is their macromolecular architecture [24,25,26,27]. Controlled synthesis of BC is promising for the creation of new supramolecular architectures with very specific self-assembly behavior [28,29,30,31,32,33]. Advances in the synthesis of block copolymers are creating new opportunities for unusual self-assembly processes in both bulk and thin-film materials. [19,34,35,36]. Nanoporous BCs are promising as optically transparent substrates for the test methods for the qualitative and quantitative determinations of various analytes [37,38]. One of the ways to control the diffusion properties of BC is based on the directed influence on their supramolecular organization.

In [39,40,41], microporous optically transparent block copolymers (OBC) were synthesized and studied based on block copolymers of propylene and ethylene oxides (PPEG) and 2,4-toluene diisocyanate (TDI). Reaction conditions were established, under which coplanar O-polyisocyanate blocks [39] were formed in OBC.

In [42], to improve the microphase separation, OBCs were modified with stable-to-self-condensation Cu(II)-coordinated polyoxyethylene-substituted silicas (ASiP-Cu). The synthesis of ASiP was carried out using tetraethoxysilane (TEOS) as the main reagent, polyoxyethylene glycol (PEG) was used to create branches in the structure of the resulting silicas, and copper chloride (CuCl_2_) was used as a catalyst and complexing agent. It was found that the effect of CuCl_2_ on the topological structure of ASiP-Cu depended on its content in the reaction system. Thus, when copper chloride is used in small quantities, ASiP (ASiP-Cu-0.01) is formed, characterized by a silica core of cubic topology. At the same time, CuCl_2_ in the synthesis of ASiP-Cu performs only the function of a catalyst. An increase in the CuCl_2_ content in the reaction system leads to the formation of silicas of linear topology in the ASiP composition (ASiP-Cu-0.1). As the CuCl_2_ content increases further, Cu(II) ions enter into coordination interactions with polyoxyethylene branches to form ASiP-Cu-0.5 (Figure 1).

Currently, more and more works are being published aimed at the surface modification of silicon dioxide particles with oligomers [43,44,45,46]. The most promising synthesis of silicon dioxide nanoparticles is a method based on the use of alkoxysilanes and polyoxyethylene glycols for subsequent modification of their surfaces [47,48,49,50,51,52,53]. To obtain silica using alkoxysilanes, sol–gel synthesis processes are used, which are based on the stages of hydrolysis of alkoxysilanes and the subsequent condensation of silanol groups [53,54,55,56,57,58,59]. The main problem of obtaining organo-substituted silicon dioxide nanoparticles is to make them stable to self-condensation. To solve this urgent problem, it Is necessary to select a catalytic reagent that would promote the simultaneous formation of silicas and the addition of organic, including oligomeric, branches [59,60,61,62].

In this work, the yield and strengthening of the structure of O-polyisocyanate blocks in OBC, obtained on the basis of PPEG, TDI, and AsiP-Cu, were studied depending on the content of copper chloride used in the synthesis of AsiP-Cu. The effect of AsiP-Cu-0.5 on the supramolecular structure of OBC was studied. The analytical system was obtained using polyethylene terephthalate film as a substrate for a selective layer of OBC and using PAN as an analytical reagent, and it was tested to the find the Cu(II) ions.

For practical use, the creation of test systems based on a polymer membrane obtained using irrigation technology is a difficult task. To create thin films of uniform thickness, the technology of applying a selective layer to a specially selected substrate is usually used. In this work, polyethylene terephthalate film was used as a substrate. The OBC applied to the polyethylene terephthalate substrate was prepared using 0.5 wt.% AsiP-Cu-0.5. The thickness of the OBC on the substrate surface was 10 µm. The substrate with a selective layer of the OBC (POBC) was then immersed in the PAN and PHENAZO solutions, which were adsorbed onto both the OBC and the polyethylene terephthalate substrate. Then, the resulting polymer composition was kept for 24 h in water. In this case, the PAN and PHENAZO were completely desorbed from the polyethylene terephthalate substrate of the POBC but remained in the microporous structure of the selective OBC layer. The POBC obtained in this way was further used to determine Cu(II) and magnesium ions. Increasing the efficiency of OBC as a selective layer of POBC was considered.

## 2. Materials and Methods

### 2.1. Materials

Information about the materials used can be found in [42].

### 2.2. Procedure for the Block Copolymers as Test Systems Synthesis

OBC was synthesized in a conical flask equipped with a reflux condenser. PPEG (10 g), toluene (79 mL), bisphenol A (0.04 g), acetic acid (100 μL), and the calculated amount of Cu(II)-coordinated polyoxyethylene-substituted silicas (AsiP-Cu-(0–0.5 wt.%)) were used. The reaction solution was stirred on a magnetic stirrer without heating. The procedure for the synthesis of Cu(II)-coordinated polyoxyethylene-substituted silicas is given in [42]. After the complete dissolution of the reagents in toluene, TDI (5.4 g) was introduced into the reaction solution. Then, the reaction solution was stirred for 10 min without heating.

To obtain OBC as a test system, the reaction solution was poured into Petri dishes. The thickness of the OBC obtained in a Petri dish was 100 μm. To obtain a test system on an optically transparent polymer substrate (POBC), an OBC layer was applied to a polyethylene terephthalate substrate. The OBC thickness on the substrate surface was 10 μm. 

Polyethylene terephthalate PET-A (APET) Novattro, Russia, Moscow was used to obtain POBC. Sheet thickness was 0.3 mm, density was 1.33 kg/cm^2^, operating temperature range was −40 °C–+60 °C, water absorption was 0.1–0.3, and the light transmission coefficient 88%.

Curing of OBC in both cases was carried out at room temperature for 72 h.

### 2.3. Samples Preparation

A solution of 1-(2-pyridylazo)-2-naphthol (PAN) with a concentration of 10^−3^ mol/L and 3,3′-dinitro-4,4′-bis(4-hidroxyphenilazo)biphenyl (PHENAZO) with a concentration of 10^−4^ mol/L was prepared by dissolving PAN in ethanol. Immobilization of PAN and PHENAZO on POBC was carried out by adsorption from solutions in a static mode. POBCs were kept in the PAN solution for 4 min and in the PHENAZO solution for 10 min. After drying, the polymer film materials were kept in distilled water for 24 h, during which the water was replaced three times. The POBCs obtained in this way was further used to determine copper and magnesium ions.

Solutions of CuSO_4_ and MgCl_2_ were prepared by dissolving the corresponding weights of copper(II) sulfate pentahydrate and magnesium chloride hexahydrate in distilled water. The sorption process was carried out in a static mode with periodic stirring for one hour. The determination of metal ions using PAN was carried out at pH = 7 and when using PHENAZO at pH = 9–10.

### 2.4. Measurements

#### 2.4.1. Infrared (IR) Spectroscopy

The IR spectra of the interaction products in thin films were recorded on a Nicolet iS20 FT-IR Fourier spectrometer (Thermo Fisher Scientific, Waltham, MA, USA) using frustrated total internal reflection spectroscopy. The spectral resolution was 2 cm^−1^, and the number of scans was 16.

#### 2.4.2. Electronic Spectroscopy

Electronic spectra were performed with the double-beam spectrophotometer Specord 210 plus (Analytik Jena, Jena, Germany) in thin polymer films in the spectral range from 300 to 700 nm. Scan step was 1 nm, and scan speed was 10 nm/s.

#### 2.4.3. Tensile Stress–Strain Measurements

Tensile stress–strain tests were obtained for 40 mm × 15 mm film samples using an Inspekt mini universal testing machine (Hegewald&Peschke Meß- und Prüftechnik GmbH, Nossen, Germany) at 293 ± 2 K, 1 kN, and a speed of 50 mm/min. The test was carried out until the destruction of the sample.

#### 2.4.4. Temperature Dependence of the Dielectric Loss Tangent (DLT) Measurement

The temperature dependence of the DLT of polymer samples was recorded at a frequency of 1 kHz using a measuring cell placed in a Dewar vessel filled with nitrogen. An E7–20 RLC meter and a B7–78 universal voltmeter, which performs the function of a precision thermometer, were connected to the measuring cell. The measurements were carried out in the process of increasing the temperature from −120 °C to +140 °C.

## 3. Results

### 3.1. Effect of AsiP-Cu on the OBC Structure

Since in this work, OBC was used as a polymer substrate for obtaining test systems, we studied the effect of AsiP-Cu on the completeness of the reaction of the opening of isocyanate groups at the carbonyl initiated by terminal potassium alcoholate groups of PPEG. The course of the reaction according to a given mechanism is important, since the main structural elements of OBC are rigid O-polyisocyanate and flexible polyether blocks. Previously [39], it was established that the completeness of the opening of isocyanate groups at the carbonyl and the formation of O-polyisocyanate blocks are significantly influenced by the conditions of the reaction process. To exclude the depolymerization and additional stabilization of the O-polyisocyanate blocks, conditions were created for the interaction of isocyanate groups of the *ortho* position of TDI in the already formed O-polyisocyanate blocks. For this purpose, at the final stage of TDI polymerization, an amount of water equimolar to isocyanate groups and triethylamine (TEA) as a catalyst were introduced into the reaction system. It should be noted that the creation of low temperatures and the controlled introduction of water and TEA require additional operations and generally complicate the synthesis of OBC.

It turned out that the use of modifiers makes it possible to synthesize OBC at ordinary temperatures and without the additional kinetically controlled introduction of water and TEA into the reaction system. To analyze the completion of the reaction of the formation of O-polyisocyanate blocks, IR spectroscopy is the most informative.

Figure 2 shows a diagram of possible pathways for the opening of isocyanate groups during the interaction of PPEG with a large molar excess of TDI, initiated by the terminal potassium alcoholate groups of PPEG. The opening of N = C = O groups by the N = C component leads to the formation of polyisocyanates of amide nature. However, under normal conditions and in the absence of stabilization, polyisocyanates of amide nature cyclize and form isocyanurates (structure I). Further reactions involving unreacted isocyanate groups of the ortho-position TDI lead to the formation of polyisocyanurate structures. The unusual opening of N = C = O groups with the participation of the C = O component is due to the creation of reaction conditions described in [39]. In this case, polyisocyanates of acetal nature are formed (O-polyisocyanates, structure II), the stabilization of which requires the formation of urea with the participation of isocyanate groups of TDI ortho-position that have not reacted. In addition, the formation of small amounts of urethane groups must be taken into account.

Carbonyl, as a part of the isocyanurates, is characterized by intense analytical bands in the regions of 1700 and 1410 cm^−1^ (Figure 3). The carbonyl of the urethane group is characterized by bands at 1710 cm^−1^ (associated urethane groups) and 1730 cm^−1^ (unassociated urethane groups). Urea is characterized by a low-intensity band in the region of 1620–1640 cm^−1^, which is related to the stretching vibrations of C = O bonds. The analytical band of the N = C bond in the O-polyisocyanate blocks in this case is low-intensity and is in the region of 1650–1670 cm^−1^. 

When analyzing the IR spectra of OBC, it should be taken into account that when the -N = C = O groups open to the carbonyl (C = O) component, the intensity of the bands in the region of 1700 cm^−1^ should be low. The lower the intensity, the lower the probability of isocyanurate formation and, accordingly, the higher the yield of O-polyisocyanate blocks. According to Figure 3, in the IR spectrum of unmodified OBC, the intensity of the bands in the regions of 1700 and 1410 cm^−1^ turned out to be the highest relative to the intensity of the band in the region of 1656 cm^−1^ [63]. Modification using Cu(II)-coordinated polyoxyethylene-substituted silicas led to a decrease in the intensity of the bands in the regions of 1700 and 1410 cm^−1^. AsiP-Cu-0.5 had the most significant effect on the completeness of opening of the C = O bond of the isocyanate group. Thus, for OBC modified with AsiP-Cu-0.5, an increase in the intensity of the band in the region of 1639 cm^−1^ was observed, accompanied by a decrease in intensity in the region of 1700 cm^−1^. That is, the use of AsiP-Cu-0.5 in the synthesis of OBC creates the most favorable conditions for the formation of urea, which, in accordance with the chemical structure of O-polyisocyanate blocks, is necessary for their stabilization in the structure of OBC. This, in turn, means that the use of AsiP-Cu makes it possible to increase the yield of O-polyisocyanate blocks and to carry out the interaction of PPEG with TDI under normal temperature conditions without resorting to cooling the reaction system, thereby simplifying the procedure for their synthesis. 

### 3.2. The Supramolecular Structure Study

To study the supramolecular structure of the AsiP-Cu, the temperature dependences of the dielectric loss tangent (Figure 4) and stress–strain curves (Figure 5) were obtained.

Measurements of the dielectric loss tangent (tgδ) made it possible to establish the temperature range of the onset of segmental mobility (α-transition) and the onset of mobility of the side methyl groups (β-transition) of the flexible-chain component of the OBC. Reducing the temperature of the α-transition for OBC obtained using 0.5 wt.% AsiP-Cu-0.5 (Figure 4, curve 2) from −50 °C to −55 °C and the β-transition from −100 °C to −115 °C in comparison with the unmodified OBC (Figure 4, curve 1) were consequences of the increased processes of microphase separation with the participation of the flexible-chain polyether component. In addition, for OBC obtained using 0.5 wt.% AsiP-Cu-0.5, the high-temperature transition was noticeably more pronounced in comparison with the unmodified OBC and began at a temperature of about 0 °C. Such a sharp increase in tgδ values for the modified sample compared to the unmodified one was due to a decrease in intermolecular interactions of both physical and chemical nature [64]. 

The results of the dielectric tests were consistent with the patterns of changes in the nature of the stress–strain curves (Figure 5) and confirmed the increased release of the flexible-chain component when using a modifier from AsiP-Cu-0.01 to AsiP-Cu-0.5, and when increasing its content to 0.5 wt.%.

Thus, according to Figure 5, when using a modifier from ASiP-Cu-0.01 to ASiP-Cu-0.5 and increasing its content to 0.5 wt.%, there was a significant increase in the elongation at break values. Replacing ASiP-Cu-0.01 with ASiP-Cu-0.1 and ASiP-Cu-0.5 led to an increase in elongation at break by 3–4 times. At the same time, an increase in the modifier content to 0.5 wt.% led to a noticeable decrease in the tensile strength of OBC. In all the cases studied, the deformation was not plastic but had all the signs of highly elastic stretching. Considering the high Young’s modulus reaching 120–150 MPa, the resulting polymers can be classified as high-modulus elastomers.

It was suggested that the effect of ASiP-Cu-0.5 on the microphase separation of OBC was due to the common features of the chemical structure of OBC and Cu(II)-coordinated polyoxyethylene-substituted silicas. Thus, the flexible-chain component of OBC contained polyoxyethylene fragments associated with O-polyisocyanate blocks. As a result, Cu(II) ions became a link between ASiP-Cu-0.5 and OBC. The long-range order effects that arose in this case affected the supramolecular structure of the OBC.

### 3.3. The Analytical Test Systems Obtained Using the OBC

One of the promising areas in which optically transparent microporous polymers can be successfully used is test methods for determining various substances. In practice, there is a need to use test systems away from laboratories. In this case, sensitive elements on optically transparent polymer substrates create the possibility of convenient visual observation of color changes that does not require hardware. Potential consumers of analytical test systems include engineering and research companies and clinical diagnostic laboratories. Increasing attention is being paid to analytical methods suitable for use directly at the sampling site. Such methods should be highly sensitive and selective and should be combined with rapidity, simplicity, and low cost [6,7,8,9]. The bases of chemical test methods for metal cations are the complexation reactions of organic chromophores with ions of the metal being determined, accompanied by a change in the color of the reaction system. 

In this study, the OBC was used to prepare analytical test systems, 1-(2-Pyridylazo)-2-naphthol (PAN) was used as an organic chromophore, and water-soluble copper salt was used as an analyte. According to previous studies [42], when modifying OBC using ASiP-Cu, a significant increase in the sorption capacity of the microporous block copolymer was observed. It was also found that the sorption capacity of OBC was also influenced by the method of preparing modifiers and its content in the polymer. The use of ASiP-Cu-0.5 had the most significant effect on OBC. Initially, based on the OBC and the OBC modified by the AsiP-Cu-0.5, by immobilizing the analytical reagent PAN onto the polymer, a test system was obtained. A peculiarity of carrying out an analytical reaction on a polymer membrane is the preliminary immobilization of an organic reagent (OR). The immobilization time of the OR is determined by the achievement of a constant intensity of the UV–Vis spectrum of the OR. Then, in order to avoid desorption of the OR and its capture of the analyte directly in the solution, the polymer with the OR immobilized on it is first kept in water for 24 h. 

Figure 6 shows the UV–Vis spectra of the PAN and the PAN complexes, with Cu(II) ions immobilized on the OBC and the OBC prepared using 0.5 wt.% ASiP-Cu-0.5. It turned out that when using unmodified OBC, the intensity of the PAN spectrum decreased, but a low-intensity band appeared in the region of 580 nm. When the OBC obtained using 0.5 wt.% ASiP-Cu-0.5 was used as a polymer membrane, a significant change in the shape of the spectrum and an increase in the intensity of the UV–Vis spectrum in the region of 580 nm were observed.

A comparison of Figure 6 and Figure 7 allows us to conclude that the UV–Vis spectra of the PAN and the products of complexation of the PAN with Cu(II) ions immobilized on the OBC and Cu(II) ions immobilized on the POBC showed noticeable differences. For the UV–Vis spectra of the products of complexation of the PAN with Cu(II) ions obtained on the POBC (Figure 7), a more noticeable change in the shape of the UV–Vis spectra was observed in comparison with the UV–Vis spectra of the products of complexation of the PAN with Cu(II) ions obtained on the OBC (Figure 6). Important for practical applications is the well-defined electronic spectrum for the PAN complexes with Cu(II) ions immobilized on the POBC, which was obtained with a CuSO_4_ content of 0.001 g/dm^3^ in the analytical solution.

The OBC films obtained by casting in the Petri dishes reached a thickness of 100 µm, and for films cast on the polyethylene terephthalate substrate, a thickness of 10 times less was achieved. In thin reaction layers, the moisture contained in the surrounding air made a noticeable contribution to the chemical processes of isocyanates. As was shown in the IR spectra (Figure 3), when ASiP-Cu-0.5 was used as a modifier, the highest content of urea and the lowest content of carbonyl groups were observed in the OBC structure. As a result, the highest completeness of conversion of TDI into O-polyisocyanate blocks and their stability in the composition of OBC became possible. 

To explain the observed modifying effect of the ASiP-Cu-0.5 on the properties of the selective ROBC layer, it should be taken into account that the silicas in the composition of ASiP-Cu-0.5 were linear, and Cu(II) ions coordinated the polyoxyethylene branches of ASiP-Cu-0.5 (Figure 1). Apparently, the presence of coordinated Cu(II) ions in the composition of ASiP-Cu-0.5, along with the modifying effect, had an additional catalytic effect on the reaction of urea formation between O-polyisocyanates.

Another perspective of using the OBC deposited on a polyethylene-terephthalate substrate was the determination of Mg(II) ions from aqueous solutions. In analytical laboratories, it is very often necessary to determine Mg(II) ions, which can be contained in a variety of industrial and natural materials. In most cases, the qualitative and quantitative determination of Mg(II) ions is a difficult task, since most methods are characterized by low selectivity. In the spectrophotometric analysis of Mg(II) ions, the 3,3′-dinitro-4,4′-bis(4-hidroxyphenilazo)biphenyl (PHENAZO) reagent had high selectivity and sensitivity [65]. PHENAZO in an alkaline environment formed a blue-violet adsorption compound with Mg(II) ions, and the solution of the reagent itself was colored crimson. The absorption maxima of the reagent and its complex with Mg(II) ions were observed at 490 and 560 nm, respectively. When determining Mg(II) ions using PHENAZO in aqueous solutions, the color of the resulting compound was stable for 60–80 min.

The UV–Vis spectra of PHENAZO and its complexes with Mg(II) ions adsorbed on POBC are shown in Figure 8. The sensitivity limit of the reaction of complexation of PHENAZO with Mg(II) ions in this case was 10^−4^ g/dm^3^. It should be noted that thanks to the use of PHENAZO sorbed on POBC for the analytical determination of Mg(II) ions, the color of the resulting compound was stable for a long time.

## 4. Conclusions

The effect of stable-to-self-condensation Cu(II)-coordinated polyoxyethylene-substituted silicas on the microporous block copolymers (OBCs) obtained using PPEG and TDI was studied. A feature of the interaction of TDI with PPEG was the opening of *para*-isocyanate groups through the carbonyl and the formation of coplanar O-polyisocyanate blocks initiated by the terminal potassium alcoholates of PPEG. The use of ASiP-Cu-0.5 made it possible to increase the yield and strengthen the structure of O-polyisocyanate blocks by increasing the content of urea formed from the isocyanate groups of the *ortho*-position of TDI. 

It was suggested that the effect of ASiP-Cu-0.5 on the microphase separation of OBC was due to the common features of the chemical structures of OBC and Cu(II)-coordinated polyoxyethylene-substituted silicas. Thus, the flexible-chain component of OBC contained polyoxyethylene fragments associated with O-polyisocyanate blocks. As a result, Cu(II) ions became a link between ASiP-Cu-0.5 and OBC. The long-range order effects that arose in this case affected the supramolecular structure of the OBC.

To create analytical sensors based on microporous block copolymers, test systems (POBC) were obtained by applying a selective OBC layer to the surface of a polyethylene terephthalate film. It was established that the electronic spectra of the analytical reagent PAN and the products of complexation of PAN with Cu(II) ions immobilized on the OBC and POBC showed noticeable differences. For the electronic spectra of the products of complexation of the PAN with Cu(II) ions obtained on the POBC, a more noticeable change in the shape of the spectra was observed in comparison with the spectra of the products of complexation of the PAN with Cu(II) ions obtained on the OBC. Important for practical application is the well-defined electronic spectrum for the PAN complexes with Cu(II) ions immobilized on the POBC, which was obtained with the Cu(II) ion contents in the analytical solution of 0.001 g/dm^3^. POBC was also used for the determination of Mg(II) ions from aqueous solutions using the PHENAZO reagent. The sensitivity limit of the reaction of complexation of PHENAZO with Mg(II) ions was 0.0001 g/dm^3^. 

## Figures and Tables

**Figure 1 materials-16-06810-f001:**
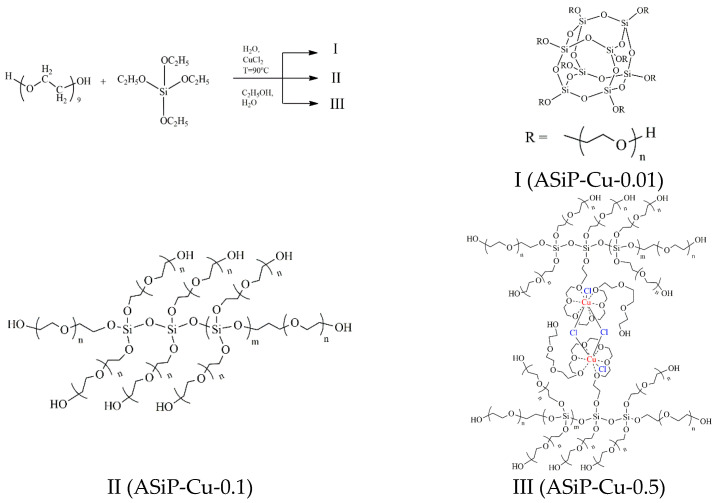
Scheme of polysiloxanes with polyoxyethylene branch formations. ASiP-Cu structure of silica cubic topology (**I**), AsiP-Cu structure of silica linear topology (**II**), and AsiP-Cu structure of silica linear topology with polyoxyethylene glycol branches coordinated by copper ions (**III**).

**Figure 2 materials-16-06810-f002:**
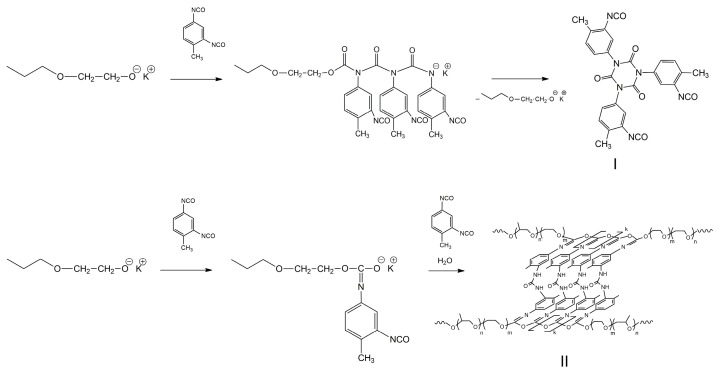
Scheme of the PPEG and TDI interaction.

**Figure 3 materials-16-06810-f003:**
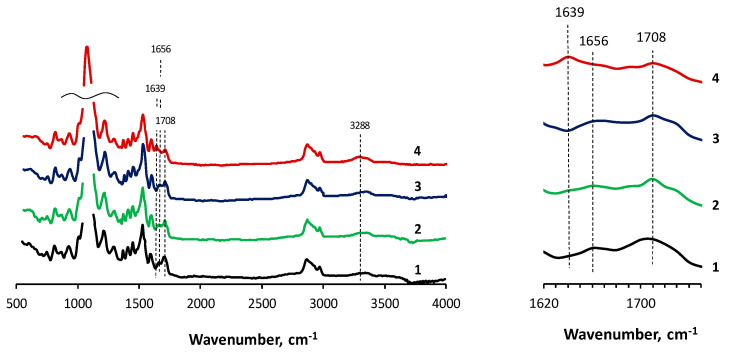
IR spectra of the OBC (1) and the OBC modified with 0.5 wt.% of AsiP-Cu-0.01 (2), AsiP-Cu-0.1 (3), and AsiP-Cu-0.5 (4).

**Figure 4 materials-16-06810-f004:**
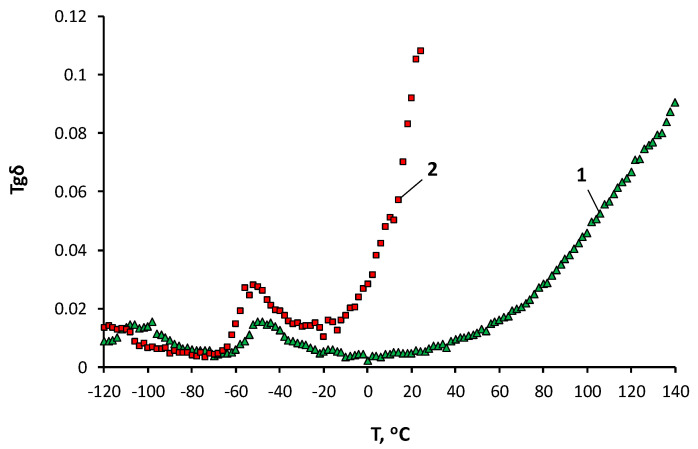
Temperature dependences of the dielectric loss tangent for the OBC obtained, with the contents of AsiP-Cu-0.5 being 0.0 (1) and 0.5 (2) wt.%.

**Figure 5 materials-16-06810-f005:**
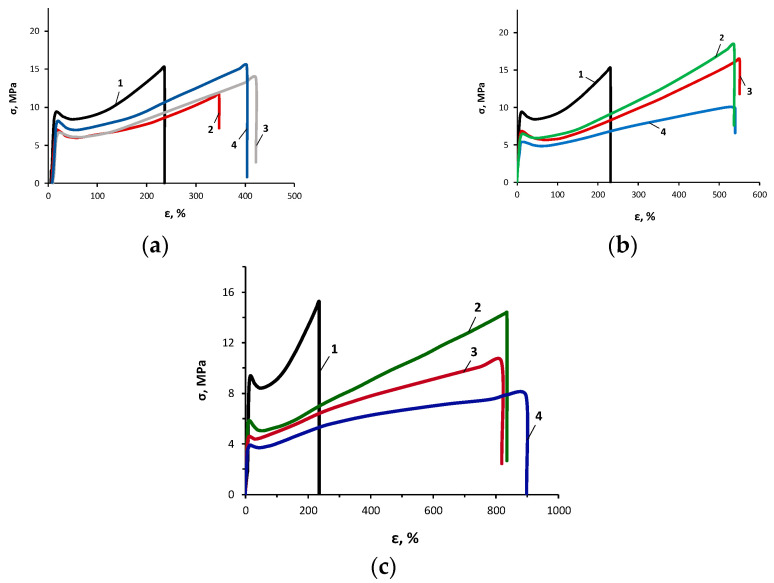
Stress–strain tests for the OBC obtained using the ASiP-Cu-0.01 (**a**), ASiP-Cu-0.1 (**b**), and ASiP-Cu-0.5 (**c**) with ASiP-Cu contents of 0 (1), 0.15 (2), 0.3 (3), and 0.5 (4) wt.%.

**Figure 6 materials-16-06810-f006:**
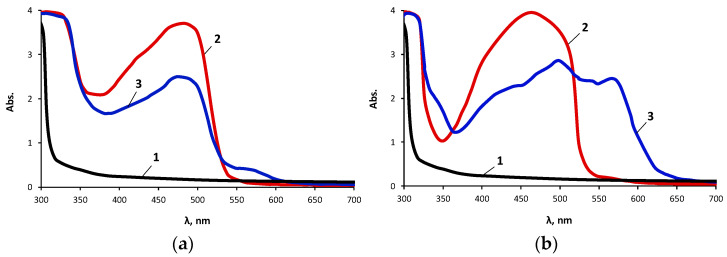
UV–Vis absorption spectra of OBC (1), of PAN, immobilized on OBC (2), and PAN complexes with Cu(II) ions immobilized on the OBC (3) (**a**), and of OBC, obtained using 0.5 wt.% of the ASiP-Cu-0.5 (1), of PAN, immobilized on OBC obtained using 0.5 wt.% of the AsiP-Cu-0.5 (2), and PAN complexes with Cu(II) ions immobilized on the OBC obtained using 0.5 wt.% of the ASiP-Cu-0.5 (3) (**b**). CuCl_2_ concentration was 0.1 g/dm^3^.

**Figure 7 materials-16-06810-f007:**
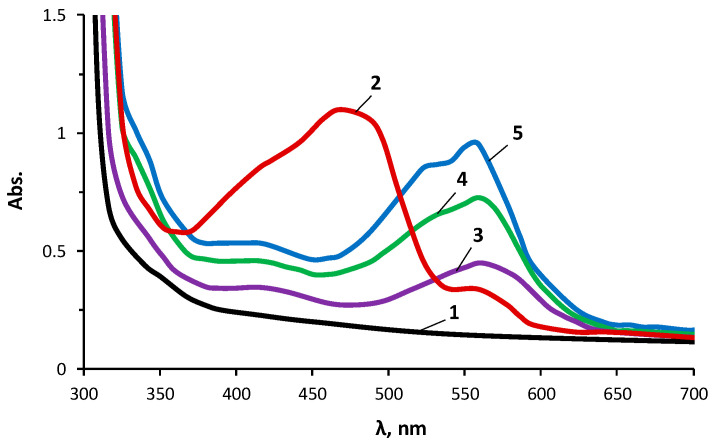
UV–Vis absorption spectra of POBC (1), POBC with immobilized PAN (2), and PAN complexes with Cu(II) ions immobilized on POBC. CuSO_4_ concentrations were 0.001 (3), 0.01 (4), and 0.1 (5) g/dm^3^.

**Figure 8 materials-16-06810-f008:**
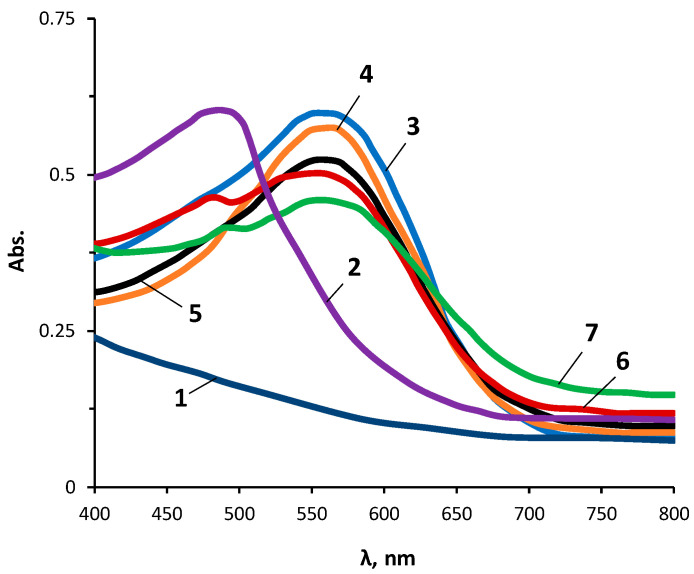
UV–Vis absorption spectra of POBC (1), POBC with immobilized PHENAZO (2), and PHENAZO complexes with Mg(II) ions immobilized on POBC: MgCl_2_ concentrations were 1 (3), 0.1 (4), 0.01 (5), 0.001 (6), and 0.0001 (7) g/dm^3^.

## Data Availability

The data presented in this study are available upon request from the corresponding author.

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
