# Peer review of "Microporous Block Copolymers Modified with Cu(II)-Coordinated Polyethylene Oxide-Substituted Silicas for Analytical Sensors"

_materials, 2023, doi:10.3390/ma16206810_

Round 1

Reviewer 1 Report

After reading the manuscript entitled: 'Microporous block copolymers modified with Cu(II)-coordinated polyethylene oxide-substituted silicas for analytical sensors" I found it very informative and I recommend the publication of this manuscript after minor English editing.

Minor English editing is needed.

Author Response

Comments and Suggestions for Authors

After reading the manuscript entitled: 'Microporous block copolymers modified with Cu(II)-coordinated polyethylene oxide-substituted silicas for analytical sensors" I found it very informative and I recommend the publication of this manuscript after minor English editing.

Response: We appreciate the reviewer's comment. English were corrected.

Reviewer 2 Report

The authors discuss the preparation of microporous block copolymers and evaluate their properties as analytical test systems. This work is seems to be a continuous research of their previous work provided in reference 42. I request the authors to take care of the following comments and revise their manuscript before publication in Materials. 
1. The originality and motivation need to be well addressed in the introduction  and indicate very clearly the difference between this work and previously published work of their group and others. 
2. I suggest revising figure 1 as it looks to be copied from the previous article of the authors!

3. The manuscript should be revised very carefully because there are number of repeated sentences; e.g page 7 (lines 216 - 218, 246-248) and sone sentences are not clear or incomplete; e.g line 159, 193-195.

4. The reaction scheme in Figure 3 is not clear for the reader and I suggest showing the abbreviation or labels of main compounds and define these labels in the caption. 

5. The results and discussion section of the manuscript lack the use of references to support the obtained findings and explanation. For example, a reference is needed in line 189, 206-207, 225-226, 241-242.

6. Line 200- do you mean curve 2 because no curve 3 in Figure 5. Please check and edit accordingly. 

7. Revise caption of Figure 7 and include spectrum 3. 

-

Author Response

Comments and Suggestions for Authors

The authors discuss the preparation of microporous block copolymers and evaluate their properties as analytical test systems. This work is seems to be a continuous research of their previous work provided in reference 42. I request the authors to take care of the following comments and revise their manuscript before publication in Materials.

Comments 1: The originality and motivation need to be well addressed in the introduction and indicate very clearly the difference between this work and previously published work of their group and others.

Response: We appreciate the reviewer's comment. Added the use of PHENAZO as an analytical reagent and the feasibility of using a polyethylene terephthalate substrate to the aim in the introduction to improve the novelty of the paper. The results of the study of PHENAZO as an analytical reagent adsorbed on OBC and ROBC have been added.

Comments 2:  I suggest revising figure 1 as it looks to be copied from the previous article of the authors!

Response: The figure 1 has been removed due to comments.

Comments 3: The manuscript should be revised very carefully because there are number of repeated sentences; e.g page 7 (lines 216 - 218, 246-248) and sone sentences are not clear or incomplete; e.g line 159, 193-195.

Response: Corrected.

Comments 4: The reaction scheme in Figure 3 is not clear for the reader and I suggest showing the abbreviation or labels of main compounds and define these labels in the caption.

Response: Corrected. Changes have been made, line 177.

Comments 5: The results and discussion section of the manuscript lack the use of references to support the obtained findings and explanation. For example, a reference is needed in line 189, 206-207, 225-226, 241-242.

Response: References have been added to the text.

Comments 6: Line 200- do you mean curve 2 because no curve 3 in Figure 5. Please check and edit accordingly.

Response: Corrected.

Comments 7:  Revise caption of Figure 7 and include spectrum 3.

Response: Corrected.

Reviewer 3 Report

The authors submitted their manuscript entitled “Microporous block copolymers modified with Cu(II)-coordinated polyethylene oxide-substituted silicas for analytical sensors”. In this work, ASiP-Cu-0.5 makes it possible to increase the yield and strengthening of the structure of O-polyisocyanate blocks by increasing the content of urea formed from the isocyanate groups of the ortho-position of TDI. The results obtained in this article are very interesting since Test systems to analyze Cu(II) ions were obtained. However, this article is very similar to the one indicated in reference 42 carried out by some of the same authors. After reviewing the manuscript, the reviewer recommends revisions for that the manuscript is suitable for publication in materials. 

Author Response

Comments and Suggestions for Authors

The authors submitted their manuscript entitled “Microporous block copolymers modified with Cu(II)-coordinated polyethylene oxide-substituted silicas for analytical sensors”. In this work, ASiP-Cu-0.5 makes it possible to increase the yield and strengthening of the structure of O-polyisocyanate blocks by increasing the content of urea formed from the isocyanate groups of the ortho-position of TDI. The results obtained in this article are very interesting since Test systems to analyze Cu(II) ions were obtained. However, this article is very similar to the one indicated in reference 42 carried out by some of the same authors. After reviewing the manuscript, the reviewer recommends revisions for that the manuscript is suitable for publication in materials.  

Comments 1: Some parts of the manuscript must be rewritten to emphasize what is new with respect to the work done in reference 42.

Response: We appreciate the reviewer's comment. Added the use of PHENAZO as an analytical reagent and the feasibility of using a polyethylene terephthalate substrate to the aim in the introduction to improve the novelty of the paper. The results of the study of PHENAZO as an analytical reagent adsorbed on OBC and ROBC have been added.

Comments 2: Figure 1 is the same as figure 1 of reference 42. Is it necessary to put it in this work?

Response: The figure 1 has been removed due to comments.

Comments 3: Figure 2 is the same as figure 10 of reference 42. Is it necessary to put it in this work?

Response: Figure 2 is necessary to make it easier for the reader to perceive the work.

Comments 4: Figure 7 is the same as Figure 15 of reference 42. In this reference PAN complexes with CuCl2 are used while in the present work CuSO4 is used. The result is the same for both jobs.

Response: Thank you for your comment. This is our error. Indeed, copper chloride was used as an analyte in Ref. 42. In this regard, appropriate adjustments were made to the text. Copper sulfate was used to determine copper ions using OBC on a polyethylene terephthalate substrate.

Comments 5: Replace the term electronic spectra with UV-Vis absorption spectra

Response: Corrected.

Comments 6:  Figure 7. Symbol б should be changed by b. It is not indicated in the caption what UV-Visabsorption spectra 3 is.

Response: Corrected.

Comments 7: The analytical system is the same as that used in reference 42. The difference between both works is the polyethylene terephthalate film as a substrate?

Response: Added the use of PHENAZO as an analytical reagent and the feasibility of using a polyethylene terephthalate substrate to the aim in the introduction to improve the novelty of the paper. The results of the study of PHENAZO as an analytical reagent adsorbed on OBC and ROBC have been added.

Comments 8: The sentence 235 to 245 should be moved to the introduction section. This sentence is very important since it reflects the usefulness of the work done. Therefore, authors should move it to the introduction and give it more emphasis.

Response: Corrected.

Comments 9: The sentence 310 to 317 is practically the same as the abstract. To avoid this, the authors should rewrite it.

Response: In accordance with the remark, the abstract has been rewritten, so the conclusions has not been rewritten.

Reviewer 4 Report

Report on Materials-2634280

In this manuscript the authors proposed the inclusion of Cu(II)-coordinated polyoxyethylene-substituted silicas in the synthesis of microporous optically transparent block copolymers based on propylene and ethylene oxides (PPEG), and 2,4-toluylene diisocyanate. Potassium alcoholates on PPEG initiate formation on para-isocyanate groups and thus obtaining of coplanar O-polyisocyanate blocks. The properties of microporous films were characterized using infrared spectroscopy, optical transmission properties, mechanical properties, and dynamic mechanical analysis. The authors claim that the presence of Cu(II)-coordinated polyoxyethylene-substituted silicas (type ASiP-Cu-0.5) in the microporous substrate material improves its mechanical strength, and supramolecular structure, and consequently, increase its microphase separation and the sorption abilities.

The manuscript was written consistently and the work performed is of interest to the Materials journal readership because it deals with very important, and still investigated subject as an increase of stability, an improvement of supramolecular structure, and an enhancement of sorption abilities of microporous optically transparent block copolymers. Some corrections in the text and answers of arisen issues from the authors should be accomplished as mentioned below:

-1- Some sentences in the manuscript are unclear in English and should be corrected or rephrased.

-2- In the Page 6 – In the Figure 5, curve 3 is not existed, probably the Authors mean curve 2.

-3- In the Page 7, lines 217-218 are non-sense.

-4- In the Figure 7, curves 3 are not explained what they come from.

Must be improved

Author Response

Comments and Suggestions for Authors

Comments 1:  Some sentences in the manuscript are unclear in English and should be corrected or rephrased.

Response: We appreciate the reviewer's comment. Some sentences in the manuscript have been corrected.

Comments 2:  In the Page 6 – In the Figure 5, curve 3 is not existed, probably the Authors mean curve 2.

Response:  Corrected.

Comments 3:  In the Page 7, lines 217-218 are non-sense.

Response: Corrected.

Comments 4:  In the Figure 7, curves 3 are not explained what they come from.

Response: Corrected.

Round 2

Reviewer 3 Report

After reviewing the document that the authors have provided taking into account the referee's recommendations, the document is accepted in this form for publication.